# Association of lifetime e-cigarette and/or tobacco use and last year cannabis use among university students: A cross-sectional secondary analysis of a national survey

Lucero Sangster-Carrasco [ORCID], Dora Blitchtein-Winicki [ORCID]*

Universidad Peruana de Ciencias Aplicadas, Chorrillos, Lima, Peru

☯ These authors contributed equally to this work.
* dblit2007@gmail.com

## Abstract

### Background

The accelerated increase in the use of e-cigarette by university students in recent years has incremented nicotine use in addition to tobacco, but it is not known whether the use of cannabis is related to these alternatives. This study analyzes the association between the use of electronic cigarettes and/or tobacco and the use of cannabis in the last 12 months. An analytical cross-sectional study conducted based on the III Andean epidemiological study on drug consumption among university students in 2016. A multivariate analysis performed using a generalized linear family model and the log Poisson link option. The results were shown as Prevalence Ratios (PR) in raw (PRc) and adjusted (PRa) models, and the analysis also employed 95% confidence intervals (95% CI) for the percentages.

### Setting

Ten Peruvian universities.

### Participants

University students' lifetime.

### Exposure

Participants' e-cigarette and/or tobacco consumption, based on questions about consumption of tobacco and e-cigarettes. Participants were categorized into those consuming: only e-cigarettes, only tobacco, both, or neither.

### Outcome

The use of cannabis by participants in the last 12 months. Among the university students who met the inclusion criteria (n = 3981), the prevalence of using both e-cigarettes and tobacco (dual) throughout life was 11.6% (95% CI: 10.1 to 13.3), and only e-cigarettes was

**Data Availability Statement:** All relevant data are within the paper and its Supporting Information files.

**Funding:** The authors received no specific funding for this work.

**Competing interests:** The authors have declared that no competing interests exist.

1.2% (95% CI: 0.7 to 1.8). The use of cannabis in the last 12 months was 5.2% (95% CI: 4.1 to 6.5). Results of this study found a sequential probability gradient of cannabis use, where those with dual use had 58.5 times more probability of having used cannabis in the last 12 months (PRa 58.5, 95% CI: 20.9 to 163.7, p <0.001) compared to those who used none; whereas those who used only tobacco were 33.3 times more likely to have used cannabis in the last 12 months (PRa 33.3, 95% CI: 11.9 to 93.2, p <0.001), those who reported using only e-cigarette had 9.8 times more probability of having used cannabis in the last 12 months (PRa 9.8, 95% CI: 1.6 to 60.4, p = 0.014). We found an increase in the probability of having used cannabis in the last 12 months among university students who reported tobacco and/or e-cigarettes use. A sequential probability gradient was found and it showed that, in comparison to students who informed being nonusers of tobacco and/or e-cigarette, university students who were lifetime dual users were most likely to having used cannabis in the last 12 months, followed by those reporting an exclusive lifetime use of tobacco, as well as those students who reported an exclusive lifetime use of e-cigarettes. It is necessary to raise awareness of the association of tobacco and e-cigarette use with an increased possibility of using cannabis. Studies are needed in different cultural contexts to investigate the progression of electronic cigarette and/or tobacco use, as well as their dosage, intensity of use, concurrent use, and neurological and behavioral mechanisms that are related to the use of cannabis and other illegal drugs that may increase dependence and mental health problems.

## Introduction

The use of tobacco products among young people has grown in recent years and is causing a significant increase in nicotine exposure. Nicotine, the component responsible for causing dependence, triggers the release of dopamine, which activates different brain areas and stimulates addiction-related reward circuits [1]. Exposure to nicotine primarily occurs through tobacco smoking, which has prevailed for centuries. This exposure is known as the leading cause of preventable death, which often takes place when individuals inhale burning chemicals. One such method is vaping using an electronic cigarette, which has become very popular since 2004. E-cigarettes, which are portable battery-powered vaporizers that young people perceive as part of a growing technological culture [2], use heat and transform various liquid or oily substances (which contain nicotine) into aerosols that are inhaled. However, even without the chemicals the burning of cigarettes produces, the effect of nicotine from vaping can have harmful effects on the central nervous system [3].

A meta-analysis showed that, when e-cigarettes entered the market between 2013 and 2015, the prevalence of their use among young people was 16.4% in more than 13 countries, a prevalence that was highly variable among those countries and ranged from 5.9% in Italy to 62.1% in Poland [4]. Regarding Peru and despite tobacco consumption has decreased among university students from 22.6% in 2012 to 16.0% in 2016, the lifetime prevalence of e-cigarette use among young people was 12.6% in 2016 [5]. One of the reasons for this is possibly the lack of standardized rules around e-cigarettes, which enables adolescents to easily access liquids with different concentrations of nicotine and other similar substances, and to use e-cigarettes in public places without any control [6].

Furthermore, this increase in young people's usage of e-cigarettes is due, among others, to their novelty; the perception that they are socially acceptable; the positive expectations they

generate, such as gaining respect, improving the possibility of being liked by others, and being popular; their stress-relieving effects; boredom; and sensory experiences, such as enjoying smell and taste [2–7]. Additionally, they believe that e-cigarettes: are not harmful or are less harmful than common tobacco cigarettes; have pleasant smells and tastes; help to stop smoking; and have no restrictions on their use in public places where tobacco smoking is not allowed. Most adolescents and young people are neither aware that nicotine levels can be very high in these products [8], nor that, if used frequently, there is a greater risk of addiction [9].

These misperceptions–especially regarding their use safety, the content of nicotine, and their effectiveness in reducing smoking–increase the possibility of using e-cigarettes at an earlier age, even among those who never used tobacco. This early exposure makes them more susceptible to damages from the effects of nicotine [2–7].

With the use of e-cigarettes, certain patterns of psychoactive substance may emerge, such as the progression to tobacco consumption and then to illicit substances [10]. Similarly, the use of both tobacco and e-cigarettes is common among young people, which can lead to the maintenance or escalation of tobacco use and nicotine dependence [7, 11].

However, the most consumed illicit substance by youth population in Latin American countries is cannabis. In Peru, the prevalence of cannabis use among university students increased significantly from 10.5% to 16.9% in men between 2009 and 2012, and did not significantly increase in women: 5.6% to 6.6%. Simultaneously, there was a fivefold increase in annual incidence in Peru, from 0.48% (2009) to 2.78% (2012) [12]. This increase in cannabis use among Latin American university students may be associated with changes in risk perception [12]. This could be due to the legalization and flexibility of its use in some countries of the region, and the greater ease of access in recent years. Additionally, the diversification of products derived from cannabis, along with technological advances, has contributed to new methods of consumption, even though smoking continues to be the most common one [13, 14]. Furthermore, the use of cannabis has also mental health consequences because of its psychoactive effects and addictive properties, which are mainly mediated by tetrahydrocannabinol (THC), the active metabolite. The higher the concentration of THC exposure in young people is, the worse the health effects are [10]. Its regular use during adolescence is related to poor academic performance, lower achievement levels, academic dropout, predisposition to addiction, early onset of psychosis [15, 16], major depression, suicidal behavior, and suicidal ideation [17].

There are several mechanisms by which the relationship between nicotine and cannabis consumption has been explained. One mechanism might be through genetic and/or environmental characteristics that condition the higher risk of consumption and abuse of or dependence on nicotine or cannabis in some people. Another mechanism explains this relationship with faster, longer lasting effects and respiratory adaptation due to the use of the same nicotine and cannabis route of administration either by smoking or vaping. An additional mechanism might also be the predictable progression in drug use, which starts with legal drugs and is followed by illegal drugs [18]. All these explanatory mechanisms may take place either independently or concomitantly.

As per psychoactive substance abuse in Peru, the sequence might be first from legal substances, such as tobacco, at an earlier age to other illegal substances such as cannabis, which is the most consumed illegal substance and has a direct impact on the increasing probability of consuming other illegal drugs [19]. The relationship between tobacco use and the initiation, maintenance and escalation in cannabis use has been previously identified [19]. On the other hand, a recent research suggests that the use of electronic cigarettes accelerates certain patterns of psychoactive substance use, such as the progression to tobacco, and the subsequent use of cannabis and other illicit substances [10, 20]. Studies on adults, who use both e-cigarettes and

tobacco (dual) concurrently with cannabis, show that this consumption causes greater dependence on nicotine, increases severe psychiatric and neurological effects, and causes the appearance of adverse effects on life and health quality [18, 21, 22]. When gathering information on various patterns of use and variations in the use of substances in different countries and cultures, no Latin American studies that analyze the association between the use of tobacco and/or e-cigarettes and the consumption of cannabis among youth are found. Therefore, the objective of this study is to identify the association between lifetime tobacco and/or e-cigarette use and the cannabis use in the last 12 months among Peruvian university students in 2016.

## Materials and methods

### Study design

This is a cross-sectional secondary analytical study based on the III Andean epidemiological study on drug usage in the Peruvian university population in 2016 [23].

### Setting

An original study [23] that used a standardized methodology and comprised a population of students from Peruvian public and private universities in cities with 300,000 or more inhabitants, with at least 60% of urban population as participants. A random two-stage sampling was conducted. In the first stage, 12 universities were selected, but only 10 agreed to participate (total population: 145,081). In the second stage, 21,144 university students were randomly selected, 4,259 of which responded to the survey, and 4,060 met the selection criteria. The information was collected through self-reporting on a standardized survey about consumption of legal and illegal drugs and other sociodemographic variables, which was sent by email.

### Participants

Regarding the present study, 3,981 university students met the selection criteria, including those who registered answers to all the following questions: "Have you ever used cannabis in the last twelve months?," "Have you ever experienced or used electronic cigarettes?," and "Have you ever experienced or used tobacco in your life?." The exclusion criterion was that students were under 18 years of age.

### Study variables

The main independent variable was the use of e-cigarettes and/or tobacco report, which was assessed through the following yes/no questions: "Have you ever smoked cigarettes or another form of tobacco in your life?" and "Have you ever smoked e-cigarettes in your life?" Responses were categorized into only e-cigarettes, only tobacco, both, and neither. The dependent variable was the self-reporting of cannabis use in the last 12 months, which was the answer to the yes/no question "Have you used cannabis in the last twelve months?" Additionally, other variables that were taken into consideration during analysis were, among others, age in years, sex, self-reported socioeconomic level, [24] parental control, [25] alcohol consumption in the last 12 months, [26] illicit drug use, [25] age of cannabis initiation, availability of cannabis in the last 12 months, and cannabis exposure opportunity in the last 12 months.

### Data source/measurement

This included self-reporting on the use of tobacco, e-cigarettes, cannabis, and other legal drugs, as it has proven to be a reliable measure of this behavior and is correlated with biomarkers [27].

## Study size

The power of this study was calculated using OPEN EPI version 3.01, with 95% confidence level, 30.6% prevalence of cannabis use among students who reported e-cigarette use, and 8.9% prevalence among those who did not. Furthermore, the database shows that 586 university students reported e-cigarette use, whereas 3,395 did not. It was estimated that the study power was greater than 80 percent [28].

## Statistical methods

The analysis was performed with STATA 16 MP and took into consideration a confidence level of 95%. Moreover, the weights assigned according to the complex study design were accounted for using the "svyset" commands. The descriptive results were obtained through frequencies and weighted percentages with confidence intervals for categorical variables, as well as mean and standard deviation for numerical variables.

As per the bivariate analysis of the association between tobacco and/or e-cigarettes and sociodemographic variables, as well as the relationship of these variables to the use of cannabis in the last 12 months, the Pearson's chi-square with Rao-Scott correction was used.

For the crude and adjusted multivariate analysis, a generalized linear model (GLM) and the log Poisson link option, which allowed to build the model and provided statistical measures to assess the model fit, were applied, [29] and the measurements of association were presented as a prevalence ratio (PR). This ratio was preferred over odds ratio (OR) and Logistic Regression that is commonly used in drug usage surveys analysis, as it is easier to interpret and its use prevents overestimation [30, 31]. Likewise, the collinearity of the independent variables that were entered in the adjusted model was assessed through the variance inflation factor (VIF), with a reference value of 10, and no collinearity was found. Correlation was also assessed and no value equal to or greater than 0.5 was found.

Moreover, an epidemiological criterion was considered for entering variables–age category, sex, [18] socioeconomic stratum, [32] alcohol consumption in the last 12 months, [15] and illicit drug use [15] in the adjusted models.

## Ethical considerations

In the original survey, the study coordinator at each university sent a letter explaining the study objective, information confidentiality guarantees and the corresponding safeguards to each randomly selected student in order for them to be able to respond voluntarily, safely and confidently. This letter also included the authorized internet address where the questionnaire was located, as well as a unique username and password. The information was collected through self-reporting on a standardized online survey, where their informed consent was obtained; this questionary was available on an Organization of American States (OAS) server. The exclusion criteria were students who did not agree to the informed consent and those who did not answer the compulsory questions specified in the survey, such as those referring to alcohol consumption. Confidentiality was maintained, and ethical considerations were fulfilled [23]. The Ethics Committee of the Universidad Peruana de Ciencias Aplicadas approved this study, a secondary analysis, through document CEI/112-07-19. The database analysis was coordinated with the National Commission for Development and Life without Drugs (DEVIDA, in Spanish) and the Inter-American Observatory on Drugs (OID, in Spanish). Anonymity of institutions and participants was kept.

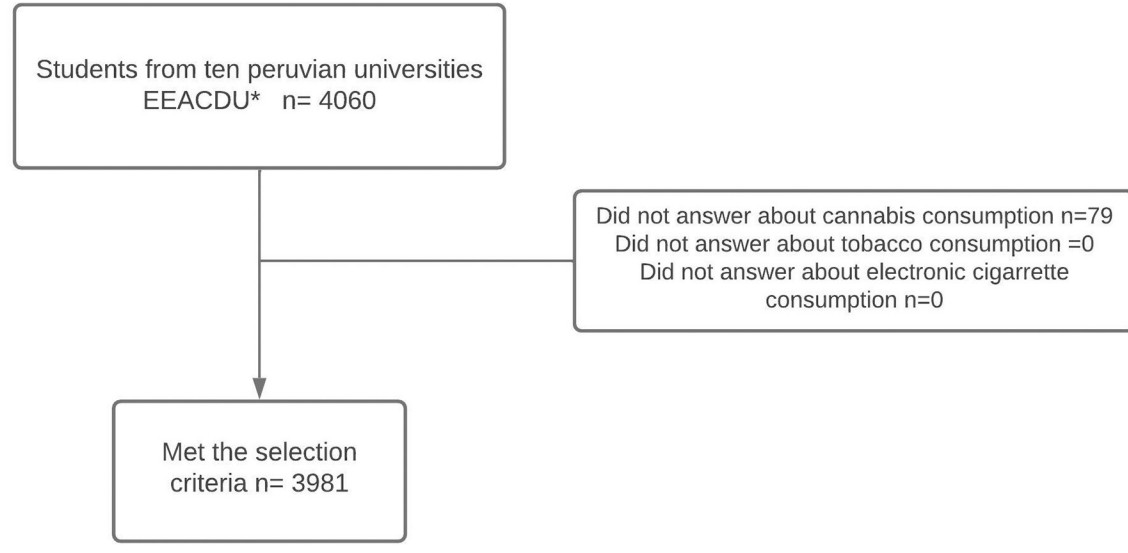

*III Andean Epidemiological Study on Drug Consumption in University Students (EEACDU), Peru 2016

**Fig 1. Study participants selection flowchart.**

## Results

As shown in Fig 1, 3,981 participants met the selection criteria for the study. Table 1 shows that 51.2% were male, 27.1% were 19–20 years old and 13.1% were 18 years old, 70.5% reported a regular socioeconomic status, and 38.1% indicated that they worked and studied.

Regarding cannabis use, 14.8% (95% CI: 13.0% to 16.7%) reported having used cannabis in their lifetime; the use prevalence in the last 12 months was 5.2% (95% CI: 4.1% to 6.5%), while, regarding tobacco use, 53.1% (95% CI: 50.5% to 55.7%) reported having used tobacco in their lifetime, and the use prevalence in the last 12 months was 29.6% (95% CI: 27.3% to 32.1%). As per e-cigarettes, 12.7% (95% CI: 11.2% to 14.5%) reported having used e-cigarettes in their lifetime.

Table 2 shows that, regarding tobacco consumption and/or e-cigarette use, 46.8% reported not having smoked either, 38.5% reported smoking exclusively tobacco, 13.2% reported having used both, and only 1.6% reported using e-cigarettes exclusively. University students who only reported e-cigarette use, also reported using cannabis for the first time at a younger age than those who reported using only tobacco (mean of 16.7 years (SD 3.7) vs. mean of 19.3 years (SD 2.4), respectively; p = 0.001).

As shown in Table 3 with respect of the association of sociodemographic characteristics with cannabis use in the last 12 months, male students reported using cannabis more than females (7.1% vs. 3.2%, respectively; p <0.001). Younger age cohorts, including 21–22, 19–20 and 18-year-old students, had a higher reporting rate of cannabis use than older cohorts that included 23–24 and >25-year old students (8.0%, 6.2% and 4.9% vs. 2.3% and 2.9%, respectively; p = 0.003). Similarly, cannabis use was higher in those university students who reported having consumed alcoholic beverages mixed with energy drinks in the last 12 months than in those who did not (13.9% vs. 5.29%, respectively; p <0.001). Students with an affirmative report of cannabis availability in the last 12 months had a higher report of cannabis use than those who denied it (12.1% vs. 2.0%, respectively; p < 0.001). Furthermore, the report of

**Table 1. Sociodemographic characteristics of Peruvian university students in 2016.**

| | | Total n = 3981 | | |
| --- | --- | --- | --- | --- |
| | | | | CI 95% |
| Characteristics | n | (%)[a] | Ll[a] | UL[a] |
| **Sex** | | | | |
| Masculine | 1762 | 51.2 | 48.5 | 53.8 |
| Femenine | 2219 | 48.8 | 46.2 | 51.5 |
| **Socioeconomic status** | | | | |
| Good or very good | 375 | 11.4 | 9.8 | 13.3 |
| Regular | 2732 | 70.5 | 68.1 | 72.8 |
| Bad or vey bad | 874 | 18.0 | 16.2 | 20.0 |
| **Age categories** | | | | |
| 18 years | 718 | 13.1 | 11.6 | 14.7 |
| 19–20 years | 1053 | 27.1 | 24.8 | 29.6 |
| 21–22 years | 889 | 23.7 | 21.5 | 26.0 |
| 23–24 years | 556 | 15.9 | 14.0 | 17.9 |
| >25 years | 765 | 20.3 | 18.2 | 22.6 |
| **Work in addition to studying** | | | | |
| No | 2496 | 61.9 | 59.2 | 64.4 |
| Yes | 1485 | 38.1 | 35.6 | 40.8 |
| **Failed subjects in his/her university life** | | | | |
| No | 1381 | 28.3 | 26.1 | 30.5 |
| Yes | 2527 | 69.8 | 67.4 | 72 |
| Does not apply | 73 | 2.0 | 1.3 | 2.9 |
| **Parental control** | | | | |
| High | 713 | 14.2 | 12.5 | 16.0 |
| Moderate | 1658 | 39.9 | 37.4 | 42.6 |
| Low | 723 | 19.5 | 17.5 | 21.7 |
| None | 887 | 26.4 | 24.0 | 28.9 |
| **Alcohol consumption in the last 12 months** | | | | |
| No | 1376 | 32.6 | 30.1 | 35.1 |
| Yes | 2605 | 67.4 | 64.9 | 69.9 |
| **Problematic alcohol use in last 12 months** | | | | |
| No | 2231 | 58.5 | 55.9 | 61.1 |
| Yes | 374 | 8.9 | 7.6 | 10.4 |
| Has not consumed alcohol in the past 12 months | 1376 | 32.6 | 30.1 | 35.1 |
| **Consumption of alcoholic beverages mixed with energy drinks in last 12 months** | | | | |
| No | 2057 | 55.6 | 52.9 | 58.2 |
| Yes | 548 | 11.8 | 10.4 | 13.5 |
| Has not consumed alcohol in the past 12 months | 1376 | 32.6 | 30.1 | 35.1 |
| **Use of tranquilizers in the last 12 months** | | | | |
| No | 3911 | 98.5 | 97.8 | 99 |
| Yes | 70 | 1.5 | 1.0 | 2.2 |
| **Cannabis use in the last 12 months** | | | | |
| No | 3755 | 94.8 | 93.6 | 95.9 |
| Yes | 226 | 5.2 | 4.1 | 6.5 |
| **Illicit drug use in life** | | | | |
| No | 3735 | 93.1 | 91.6 | 94.3 |
| Yes | 246 | 6.9 | 5.7 | 8.4 |

*(Continued)*

**Table 1.** (Continued)

| Characteristics | n | (%)ᵃ | CI 95% | |
|---|---|---|---|---|
| | | | Llᵃ | ULᵃ |
| **Age of cannabis initiation use** [b] | | | | |
| | | 18.9(2.5) | | |
| **Availability of cannabis in the last 12 months** | | | | |
| No | 2754 | 68.9 | 66.5 | 71.3 |
| Yes | 1227 | 31.1 | 28.7 | 33.5 |
| **Cannabis exposure opportunity in the last 12 months** | | | | |
| No | 3425 | 87.6 | 85.8 | 89.2 |
| Yes | 556 | 12.4 | 10.9 | 14.2 |

*Note: Total n = 3981*

[a] weighted percentage and confidence interval.

CI: Confidence Interval.

LL: Lower limit.

UL: Upper limit.

[b] mean and standad deviation (SD).

cannabis exposure opportunity was related with a higher report of cannabis use in comparison to students who did not report this exposure (12.1% vs. 2.0%, respectively; $p < 0.001$).

The crude analysis of the association between students who have used tobacco and/or e-cigarettes and cannabis in the last 12 months, and those students who used cannabis but did not use tobacco and/or e-cigarettes, was compared in Table 4. The analysis showed that those who reported e-cigarette and/or tobacco (dual) use were 107.7 times more likely to have consumed cannabis in the last 12 months (PRc 107.7, 95% CI: 40.7 to 285.1, $p <0.001$), whereas those who used tobacco (only) were 46.5 times more likely to have used cannabis in the last 12 months (PRc 46.5, 95% CI: 17.4 to 124.0, $p <0.001$), and those who reported only e-cigarette use were 14.7 times more likely (PRc 14.7, 95% CI: 2.5 to 86.8, $p = 0.003$) to have consumed cannabis during the same period of time.

However, according to the adjusted analysis (Table 4), those who used e-cigarettes and tobacco (dual) were 58.5 times more likely to have used cannabis in the last 12 months, compared to those who reported not having used either (PRa 58.5, 95% CI: 20.9 to 163.7, $p <0.001$). Moreover, those who used tobacco (only) were also found to be 33.3 times more likely to have used cannabis in the last 12 months than those who reported never having used either (PRa 33.3, 95% CI: 11.9 to 93.2, $p <0.001$). Among those who reported only e-cigarette use, they were found to be 9.8 times more likely to have consumed cannabis in the last 12 months (PRa 9.8, 95% CI: 1.6 to 60.4, $p = 0.004$), in comparison to those who used neither e-cigarettes nor tobacco.

This study identified other factors related to a higher prevalence of cannabis use in the last 12 months: illicit drug use (excluding cannabis). Those who reported using these substances were 3.1 times more likely to have consumed cannabis than those who reported not using them (PRa 3.1, 95% CI: 2.0 to 4.9, $p <0.001$).

In addition, age was associated with a lower prevalence of cannabis use in the last 12 months. Therefore, when compared to the over 25-year-old group, the 21–22, 19–20 and 18-year-old groups had 3.5 (PRa 3.5, 95% CI: 1.7 to 7.2, $p = 0.001$), 3.3 (PRa 3.3, 95% CI: 1.6 to 6.8, $p = 0.002$) and 3.1 (PRa 3.1, 95% CI: 1.5 to 6.4, $p = 0.002$) times more probability of consuming cannabis, respectively.

**Table 2. Association of sociodemographic characteristics with e-cigarette and /or tobacco use in Peruvian university students in 2016.**

| | None n = 1862 45.7% (CI95% 43.1; 48.4) | | | | Only tobacco n = 1533 41.5% (CI95% 38.9; 44.2) | | | | E-cigarettes n = 62 1.2% (CI95% 0.7; 1.8) | | | | Both n = 524 11.6% (CI95% 10.1; 13.3) | | | | $p^b$ |
|---|---|---|---|---|---|---|---|---|---|---|---|---|---|---|---|---|---|
| | | | CI 95% | | | | CI 95% | | | | CI 95% | | | | CI 95% | | |
| | n | %[a] | Ll[a] | UL[a] | n | %[a] | LL[a] | UL[a] | n | %[a] | LL[a] | UL[a] | n | %[a] | LL[a] | UL[a] | |
| **Sex** | | | | | | | | | | | | | | | | | |
| Masculine | 561 | 31.7 | 28.3 | 35.5 | 880 | 52.9 | 49.0 | 56.7 | 28 | 0.9 | 0.5 | 1.8 | 293 | 14.4 | 12.0 | 17.2 | <0.001 |
| Femenine | 1301 | 60.4 | 56.8 | 63.9 | 653 | 29.6 | 56.8 | 63.9 | 34 | 1.4 | 0.8 | 2.5 | 231 | 8.6 | 7.0 | 10.6 | |
| **Socioeconomic status** | | | | | | | | | | | | | | | | | |
| Good or very good | 380 | 45.0 | 39.4 | 50.7 | 295 | 34.6 | 29.5 | 40.1 | 28 | 2.8 | 1.4 | 5.6 | 171 | 17.6 | 13.9 | 22.0 | <0.001 |
| Regular | 1304 | 46.8 | 43.7 | 50.0 | 1079 | 41.7 | 38.5 | 44.9 | 29 | 0.8 | 0.4 | 1.5 | 320 | 10.7 | 8.9 | 12.8 | |
| Bad or vey bad | 1788 | 40.1 | 32.4 | 48.3 | 159 | 51.7 | 43.4 | 59.8 | 5 | 0.8 | 0.3 | 2.1 | 33 | 7.5 | 4.2 | 13.1 | |
| **Age categories** | | | | | | | | | | | | | | | | | |
| 18 years | 434 | 63.6 | 57.4 | 69.3 | 158 | 20.1 | 15.8 | 25.3 | 22 | 2.9 | 1.2 | 7.0 | 104 | 13.4 | 9.9 | 17.9 | <0.001 |
| 19-20 years | 525 | 53.3 | 48.2 | 58.4 | 341 | 31.9 | 27.3 | 36.8 | 15 | 0.9 | 0.5 | 1.6 | 172 | 13.9 | 10.9 | 17.6 | |
| 21-22 years | 401 | 42.3 | 37.1 | 47.7 | 354 | 44.8 | 39.4 | 50.4 | 12 | 1.3 | 0.5 | 1.6 | 122 | 11.6 | 10.9 | 17.6 | |
| 23-24 years | 224 | 41.6 | 35.1 | 48.5 | 262 | 47.4 | 40.7 | 54.2 | 7 | 1.1 | 0.4 | 3.3 | 63 | 9.9 | 6.6 | 14.6 | |
| >25 years | 278 | 31.3 | 26.0 | 37.1 | 418 | 59.9 | 53.8 | 65.7 | 6 | 0.3 | 0.1 | 0.7 | 63 | 8.5 | 5.6 | 12.8 | |
| **Work in addition to studying** | | | | | | | | | | | | | | | | | |
| No | 1239 | 50.4 | 47.1 | 53.7 | 865 | 36.9 | 33.7 | 40.2 | 48 | 1.0 | 0.7 | 1.4 | 344 | 11.8 | 9.9 | 13.9 | <0.001 |
| Yes | 623 | 38.2 | 34.1 | 42.4 | 668 | 49.1 | 44.8 | 53.5 | 14 | 1.4 | 0.6 | 3.4 | 180 | 11.3 | 8.9 | 14.3 | |
| **Failed subjects in his/her university life** | | | | | | | | | | | | | | | | | |
| No | 797 | 60.3 | 56.0 | 64.5 | 430 | 29.6 | 25.8 | 33.7 | 25 | 1.3 | 0.8 | 2.0 | 129 | 8.8 | 6.7 | 11.5 | <0.001 |
| Yes | 1021 | 39.2 | 36.0 | 42.4 | 1088 | 47.2 | 43.9 | 50.6 | 35 | 1.1 | 0.6 | 2.1 | 383 | 12.5 | 10.6 | 14.7 | |
| Does not apply | 44 | 68.7 | 50.3 | 82.6 | 15 | 11.2 | 5.6 | 21.1 | 2 | 0.9 | 0.2 | 3.5 | 12 | 19.3 | 0.1 | 39.4 | |
| **Parental control** | | | | | | | | | | | | | | | | | |
| High | 411 | 56.3 | 49.5 | 62.8 | 200 | 29.9 | 23.9 | 36.6 | 14 | 2.3 | 0.8 | 6.4 | 88 | 11.6 | 8.2 | 16.0 | <0.001 |
| Moderate | 840 | 51.9 | 47.8 | 56.0 | 581 | 38.2 | 34.3 | 42.4 | 24 | 0.7 | 0.5 | 1.1 | 213 | 9.1 | 7.4 | 11.2 | |
| Low | 297 | 38.9 | 33.3 | 44.7 | 309 | 44.1 | 38.3 | 50.0 | 10 | 1.8 | 0.7 | 4.7 | 107 | 15.3 | 11.3 | 20.3 | |
| None | 314 | 35.8 | 30.7 | 41.2 | 443 | 50.9 | 45.5 | 56.4 | 14 | 0.7 | 0.4 | 1.3 | 116 | 12.6 | 9.3 | 16.7 | |
| **Alcohol consumption in the last 12 months** | | | | | | | | | | | | | | | | | |
| No | 1026 | 74.6 | 70.3 | 78.4 | 281 | 21.7 | 18.0 | 25.8 | 19 | 0.7 | 0.4 | 1.2 | 50 | 3.1 | 1.9 | 5.0 | <0.001 |
| Yes | 836 | 31.8 | 28.9 | 34.9 | 1252 | 51.1 | 47.9 | 54.4 | 43 | 1.4 | 0.8 | 2.3 | 474 | 15.7 | 13.6 | 18.0 | |
| **Problematic alcohol use in last 12 months** | | | | | | | | | | | | | | | | | |
| No | 810 | 35.7 | 32.4 | 39.1 | 1039 | 49.2 | 45.7 | 52.7 | 39 | 1.5 | 0.9 | 2.6 | 343 | 13.6 | 11.5 | 16.1 | <0.001 |
| Yes | 26 | 6.3 | 3.5 | 11.0 | 213 | 64.1 | 56.2 | 71.3 | 4 | 0.5 | 0.2 | 1.4 | 131 | 29.2 | 22.5 | 36.9 | |
| Has not consumed alcohol in the past 12 months | 1026 | 74.6 | 70.3 | 78.4 | 281 | 21.7 | 18.0 | 25.8 | 19 | 0.7 | 0.4 | 1.2 | 50 | 3.1 | 1.9 | 5.0 | |
| **Consumption of alcoholic beverages mixed with energy drinks in last 12 months** | | | | | | | | | | | | | | | | | |
| No | 769 | 35.9 | 32.5 | 39.4 | 972 | 50.7 | 47.1 | 54.4 | 33 | 1.2 | 0.7 | 2.0 | 283 | 12.3 | 10.1 | 14.8 | <0.001 |
| Yes | 67 | 12.8 | 8.8 | 18.4 | 280 | 53.1 | 46.2 | 59.9 | 10 | 2.4 | 0.7 | 7.6 | 191 | 31.7 | 25.8 | 38.2 | |
| Has not consumed alcohol in the past 12 months | 1026 | 74.6 | 70.3 | 78.4 | 281 | 21.7 | 18.0 | 25.8 | 19 | 0.7 | 0.4 | 1.2 | 50 | 3.1 | 1.9 | 13.3 | |
| **Use of tranquilizers in the last 12 months** | | | | | | | | | | | | | | | | | |
| No | 1842 | 45.9 | 43.3 | 48.6 | 1496 | 41.4 | 38.8 | 44.1 | 61 | 1.2 | 0.7 | 1.8 | 512 | 11.5 | 10.0 | 13.2 | 0.549 |
| Yes | 20 | 34.7 | 17.9 | 56.4 | 37 | 47.9 | 28.9 | 67.6 | 1 | 1,3 | 0.2 | 8.8 | 12 | 16.1 | 8.0 | 29.6 | |
| **Cannabis use in the last 12 months** | | | | | | | | | | | | | | | | | |
| No | 1857 | 52.8 | 50.0 | 55.7 | 1420 | 38.1 | 35.3 | 40.9 | 60 | 1.3 | 0.8 | 2.1 | 418 | 7.8 | 6.4 | 9.4 | |
| Yes | 5 | 4.7 | 2.5 | 8.7 | 113 | 61.5 | 55.0 | 67.6 | 2 | 0.3 | 0.1 | 0.8 | 106 | 33.5 | 27.7 | 39.8 | <0.001 |
| **Illicit drug use in life (not including cannabis)** | | | | | | | | | | | | | | | | | |

(*Continued*)

**Table 2.** (Continued)

| | None n = 1862 45.7% (CI95% 43.1; 48.4) | | | | Only tobacco n = 1533 41.5% (CI95% 38.9; 44.2) | | | | E-cigarettes n = 62 1.2% (CI95% 0.7; 1.8) | | | | Both n = 524 11.6% (CI95% 10.1; 13.3) | | | | $p^b$ |
|---|---|---|---|---|---|---|---|---|---|---|---|---|---|---|---|---|---|
| | | | CI 95% | | | | CI 95% | | | | CI 95% | | | | CI 95% | | |
| | n | %[a] | LI[a] | UL[a] | n | %[a] | LL[a] | UL[a] | n | %[a] | LL[a] | UL[a] | n | %[a] | LL[a] | UL[a] | |
| No | 1770 | 46.3 | 43.5 | 49.0 | 1433 | 41.3 | 38.6 | 44.1 | 58 | 1.2 | 0.7 | 1.9 | 474 | 11.2 | 9.7 | 13.0 | 0.203 |
| Yes | 92 | 38.7 | 29.4 | 48.8 | 100 | 44.4 | 34.8 | 54.4 | 4 | 0.8 | 0.3 | 2.3 | 50 | 16.2 | 10.6 | 24.0 | |
| **Age of cannabis initiation use(mean SD)[c]** | | | | | | | | | | | | | | | | | |
| | | 17.0 (2.4) | | | | 19.3 (2.4) | | | | 16.7 (3.7) | | | | 18.4 (2.6) | | | 0.001[d] |
| **Availability of cannabis in the last 12 months** | | | | | | | | | | | | | | | | | |
| No | 1556 | 54.9 | 51.7 | 58.1 | 932 | 36.7 | 33.6 | 39.9 | 41 | 1.3 | 0.7 | 2.2 | 225 | 7.1 | 5.7 | 8.9 | <0.001 |
| Yes | 306 | 25.3 | 21.5 | 29.7 | 601 | 52.3 | 47.6 | 56.9 | 21 | 0.9 | 0.6 | 1.8 | 299 | 21.5 | 18.1 | 25.3 | |
| **Cannabis exposure opportunity in the last 12 months** | | | | | | | | | | | | | | | | | |
| No | 1778 | 50.5 | 47.6 | 53 | 1273 | 39.4 | 36.7 | 42.3 | 57 | 1.3 | 0.8 | 2.0 | 317 | 8.8 | 7.3 | 10.6 | <0.001 |
| Yes | 84 | 12.3 | 9.0 | 17 | 260 | 56.4 | 49.6 | 63.0 | 5 | 0.4 | 0.2 | 1.1 | 207 | 30.9 | 25.3 | 37.1 | |

[a]All percentages and confidence interval are weighted.

[b] Pearson's Chi square with Rao-Scott correction.

CI: Confidence Interval.

LL: Lower limit.

UL: Upper limit.

[c] mean and standad deviation (SD).

[d] Wald test

## Discussion

This study with university students revealed that there is an increasing association between lifetime patterns of tobacco and/or e-cigarette consumption and cannabis use in the last 12 months. University students who used tobacco and e-cigarettes, as well as those who used either only tobacco or only e-cigarettes, had a higher probability of using cannabis [58.5, 33.3 and 9.8 times more probability, respectively] than students who did not use either of them.

The results regarding tobacco and e-cigarette users reveal a very high probability of cannabis use, followed by a high probability among tobacco (only) users and e-cigarette (only) users. The ascending gradient of the association of cannabis use with lifetime use patterns of e-cigarette and tobacco, tobacco only and e-cigarette only use in university students differed, for instance, from findings in studies conducted in adolescents in Iceland, which identified that lifetime patterns of tobacco and/or e-cigarette use in relation to lifetime illicit substance use, including cannabis, were mostly associated with e-cigarette and tobacco use, followed by e-cigarette only and, to a lesser extent, tobacco only.

Even though the results of another study in adolescents in the United States identified the same gradual relationship of association of cannabis use with the identified patterns, the time period was last month and the prevalence of tobacco and/or e-cigarette use were very different. It found a higher prevalence of e-cigarette use only than regarding tobacco use only, which differ from our findings in the population of our study, in which e-cigarette use was much lower than tobacco use.

The advantage in the aforementioned study was the measurement about tobacco and or e-cigarette frequency of use, simultaneous use and intensity of use; however, they did not analyze lifetime use patterns [25, 32–34].

**Table 3. Association of sociodemographic characteristics with cannabis use in the last 12 months in Peruvian university students in 2016.**

| | \multicolumn Cannabis use in the last 12 months | | | | | | | | |
|---|---|---|---|---|---|---|---|---|---|
| | No | | | | Yes | | | | |
| | n = 3755 | 94.8% | | | n = 226 | 5.2% | | | |
| | IC 95% | | | | IC 95% | | | | |
| | n | (%) | Ll[a] | UL[a] | n | (%) | Ll[a] | UL[a] | p[b] |
| **Sex** | | | | | | | | | |
| Masculine | 1611 | 92.9 | 90.7 | 94.6 | 151 | 7.1 | 5.4 | 9.3 | 0.001 |
| Femenine | 2144 | 96.8 | 95.4 | 97.8 | 75 | 3.2 | 2.2 | 4.6 | |
| **Socioeconomic status** | | | | | | | | | |
| Good or very good | 808 | 93.9 | 91.2 | 95.8 | 66 | 6.1 | 4.1 | 8.8 | 0.624 |
| Regular | 2592 | 94.9 | 93.3 | 96.2 | 140 | 5.1 | 3.8 | 6.7 | |
| Bad or vey bad | 355 | 95.9 | 90.8 | 96.2 | 20 | 4.1 | 1.8 | 9.2 | |
| **Age categories** | | | | | | | | | |
| 18 years | 678 | 95.1 | 92.4 | 96.9 | 40 | 4.9 | 3.1 | 7.6 | 0.003 |
| 19-20 years | 982 | 93.8 | 90.7 | 95.9 | 71 | 6.2 | 4.2 | 9.3 | |
| 21-22 years | 823 | 92.0 | 88.1 | 94.7 | 66 | 8.0 | 5.3 | 11.9 | |
| 23-24 years | 530 | 97.7 | 96.5 | 98.6 | 26 | 2.3 | 1.5 | 3.5 | |
| >25 years | 742 | 97.1 | 93.6 | 95.9 | 23 | 2.9 | 1.6 | 5.4 | |
| **Work in addition to studying** | | | | | | | | | |
| No | 2334 | 93.8 | 91.9 | 95.2 | 162 | 6.2 | 4.8 | 8.1 | 0.013 |
| Yes | 1421 | 96.6 | 94.9 | 97.7 | 64 | 3.4 | 2.3 | 5.1 | |
| **Failed subjects in his/her university life[c]** | | | | | | | | | |
| No | 1325 | 96.4 | 94.6 | 97.6 | 56 | 3.6 | 2.4 | 5.4 | 0.065 |
| Yes | 2362 | 94.4 | 92.7 | 95.7 | 165 | 5.6 | 4.3 | 7.3 | |
| **Parental control** | | | | | | | | | |
| High | 686 | 95.0 | 90.3 | 97.5 | 27 | 5.0 | 2.5 | 9.7 | 0.058 |
| Moderate | 1579 | 95.9 | 93.9 | 97.2 | 79 | 4.1 | 2.8 | 6.1 | |
| Low | 677 | 96.1 | 94.4 | 97.4 | 46 | 3.9 | 2.7 | 5.7 | |
| None | 813 | 92.3 | 88.8 | 94.7 | 74 | 7.7 | 5.3 | 11.2 | |
| **Alcohol consumption in the last 12 months** | | | | | | | | | |
| No | 1369 | 99.2 | 97.1 | 99.8 | 7 | 0.8 | 0.2 | 2.9 | <0.001 |
| Yes | 2386 | 92.7 | 90.9 | 94.2 | 219 | 7.3 | 5.8 | 9.1 | |
| **Problematic alcohol use in last 12 months** | | | | | | | | | |
| No | 2096 | 94.1 | 92.2 | 95.6 | 135 | 5.1 | 4.4 | 7.8 | <0.001 |
| Yes | 290 | 83.4 | 77.7 | 87.9 | 84 | 2.2 | 12.1 | 22.4 | |
| Has not consumed alcohol in the past 12 months | 1369 | 99.2 | 97.1 | 99.8 | 7 | 0.8 | 0.2 | 2.9 | |
| **Consumption of alcoholic beverages mixed with energy drinks in last 12 months** | | | | | | | | | |
| No | 1931 | 94.1 | 92.1 | 95.7 | 126 | 5.9 | 4.3 | 7.9 | <0.001 |
| Yes | 455 | 86.2 | 81.6 | 89.7 | 93 | 13.9 | 10.3 | 18.4 | |
| Has not consumed alcohol in the past 12 months | 1369 | 99.2 | 97.1 | 99.8 | 7 | 0.8 | 4.1 | 6.4 | |
| **Use of tranquilizers in the last 12 months** | | | | | | | | | |
| No | 3692 | 94.9 | 93.6 | 95.9 | 219 | 5.1 | 4.1 | 6.4 | 0.326 |
| Yes | 62 | 92.1 | 82.8 | 96.6 | 7 | 7.9 | 3.4 | 17.2 | |
| **Availability of cannabis in the last 12 months** | | | | | | | | | |
| No | 2698 | 98.0 | 96.8 | 98.7 | 56 | 2.0 | 1.3 | 3.2 | <0.001 |
| Yes | 1057 | 87.9 | 84.6 | 90.5 | 170 | 12.1 | 9.5 | 15.4 | |
| **Cannabis exposure opportunity in the last 12 months** | | | | | | | | | |

(*Continued*)

**Table 3.** (Continued)

| | Cannabis use in the last 12 months | | | | | | | |
| --- | --- | --- | --- | --- | --- | --- | --- | --- |
| | No | | | | Yes | | | |
| | n = 3755 | 94.8% | | | n = 226 | 5.2% | | |
| | IC 95% | | | | IC 95% | | | |
| | n | (%) | Ll[a] | UL[a] | n | (%) | Ll[a] | UL[a] | p[b] |
| No | 3391 | 99.3 | 98.7 | 99.6 | 34 | 0.7 | 0.4 | 1.3 | <0.001 |
| Yes | 364 | 63.4 | 56.2 | 70.1 | 192 | 36.6 | 29.9 | 43.8 | |

[b] Pearson's Chi square with Rao-Scott correction.

All percentages are weighted.

CI: Confidence Interval.

LL: Lower limit

UL: Upper limit

[c] n = 68 students were not included, there were in the first semester

[d] n = 43 missing didn't answer

Furthermore, the findings about a sequential probability gradient may be related to the level of nicotine exposure. Previous longitudinal evidence has shown that adolescents and youth who use nicotine e-cigarettes may progress more rapidly to nicotine addiction and tobacco use. As aerosols inhaled through e-cigarettes release nicotine with highly oxidative free radicals, which are more addictive and more easily absorbable forms of nicotine, they cause the above progression [35, 36]. Likewise, other studies have shown that there could be a connection between the progression from e-cigarette to tobacco, as well as an increased risk of cannabis use in young people [20]. The biological mechanism may be related to the exposure to nicotine, which triggers changes in the central nervous system and leads to a stronger response to cannabis [16]. The use of both tobacco and e-cigarettes also implies that nicotine exposure is doubles and, therefore, there is a greater probability of developing nicotine dependence and progressing to cannabis use [18, 19, 37]. Other related mechanisms are comparable behaviors, such as hand-mouth movements, puffing, inhalation and exhalation, among others, which contribute to understanding the relationship between the use of e-cigarettes with nicotine and tobacco and cannabis use. These behaviors make the transition from one another a more natural process [35]. The use of tobacco and e-cigarettes together with cannabis could also be connected to risky behaviors, problematic alcohol drinking, type of personality, and the search for sensations and new experiences [34, 38].

Moreover, the association between tobacco and e-cigarette use with cannabis use could also be linked to the fact that one of the main emerging routes for cannabis use in adolescents is through vaping [10, 39]. This should be considered when reviewing the nicotine use legalization and flexibility, as well as the emerging routes regarding its use within and outside Latin America. This relationship with cannabis is also valid for those who use e-cigarettes, as one in three participants reported having used it for cannabis consumption [28]. It is known that among Hispanic American residents in various countries, cannabis use is more common than in other ethnic groups, [40] including adolescents in Mexico [26]. However, we have not been able to identify studies carried out in Peru on the prevalence and/or characteristics of this practice, although it has been proven that e-cigarettes are easily available to students on different virtual and face-to-face sales channels [5].

This study did also find an association between the pattern of using only e-cigarettes and the consumption of cannabis, unlike those studies from Canada, England, and Mexico [26, 32, 33]. These students were also younger at the time when they first used cannabis than the users

**Table 4. Crude and adjusted model of association between the use of e-cigarettes and / or tobacco in life and the use of marijuana in the last 12 months in Peruvian students in 2016.** n = 3981.

| | | IC 95% | | | | CI 95% | | |
| --- | --- | --- | --- | --- | --- | --- | --- | --- |
| | Unadjusted PR | LL | UL | p | Adjusted PR[c] | LL | UL | p |
| **Tobacco and / or electronic cigarette consumption pattern in life** | | | | | | | | |
| None | Ref. | | | | Ref. | | | |
| Only tobacco | 46.5 | 17.4 ; | 124.0 | <0.001 | 33.3 | 11.9 ; | 93.2 | <0.001 |
| Only electronic cigarette | 14.7 | 2.5 ; | 86.8 | 0.003 | 9.8 | 1.6 ; | 60.4 | 0.014 |
| Both | 107.7 | 40.7 ; | 285.1 | <0.001 | 58.5 | 20.9 ; | 163.7 | <0.001 |
| **Sex** | | | | | | | | |
| Femenine | Ref. | | | | Ref. | | | |
| Masculine | 2.2 | 1.4 ; | 3.6 | 0.001 | 1.4 | 0.9 ; | 2.3 | 0.134 |
| **Socioeconomic status** | | | | | | | | |
| Bad or very bad | Ref. | | | | Ref. | | | |
| Regular | 1.2 | 0.5 ; | 2.9 | 0.646 | 1.1 | 0.5 ; | 2.8 | 0.812 |
| Good or very good | 1.5 | 0.6 ; | 3.6 | 0.390 | 1.1 | 0.4 ; | 2.9 | 0.816 |
| **Age categories** | | | | | | | | |
| >25 years | Ref. | | | | Ref. | | | |
| 23-24 years | 0.8 | 0.4 ; | 1.7 | 0.526 | 0.9 | 0.4 ; | 2.0 | 0.840 |
| 21-22 years | 2.7 | 1.3 ; | 5.8 | 0.008 | 3.5 | 1.7 ; | 7.2 | 0.001 |
| 19-20 years | 2.1 | 1.01 ; | 4.5 | 0.045 | 3.3 | 1.6 ; | 6.8 | 0.002 |
| 18 years | 1.7 | 0.8 ; | 3.6 | 0.187 | 3.1 | 1.5 ; | 6.4 | 0.002 |
| **Alcohol consumption in the last 12 months** | | | | | | | | |
| Yes | Ref. | | | | Ref. | | | |
| No | 9.4 | 2.4 ; | 37.0 | 0.001 | 3.2 | 0.9 ; | 11.4 | 0.074 |
| **Illicit drug use (does not include canabis)** | | | | | | | | |
| No | Ref. | | | | Ref. | | | |
| Yes | 4.2 | 2.7 ; | 6.7 | <0.001 | 3.1 | 2.0 ; | 4.9 | <0.001 |

Generalized family linear model and Poisson link log option.

[c] PRa. Adjusted for: sex, socioeconomic status, age category, problematic alcohol use in the last 12 months, and use of other illicit drugs.

CI: Confidence Interval.

LL: Lower limit.

UL: Upper limit

of other nicotine patterns. Additionally, the original study assessed only the use, not the frequency, of e-cigarettes in the last 12 months, last month, or week. It also did not include questions to estimate sensation-seeking behaviors or the use of e-cigarettes by peers and/or parents. No information was collected about the sequence of use of e-cigarettes and/or tobacco, or whether they were dually consumed. The components, levels of nicotine and cannabis products used as inputs in e-cigarettes, as well as how they were used, were neither specified [23].

Therefore, this study contributes to literature in the analysis of lifetime patterns in the use of e-cigarettes and/or tobacco among young people in Latin America, since other studies have not analyzed this correlation [20, 26]. It also explores the effect of consuming alcohol and other illegal drugs, as well as parental control, which has not been frequently included before.

Lifetime prevalence of dual use of e-cigarettes and tobacco in Peruvian university students was 11.6% (95% CI: 10.1; 13.3); only tobacco was 41.5% (95% CI: 38.9; 44.2); neither tobacco nor e-cigarettes was 45.7% (95% CI: 43.1; 48.4); and only e-cigarettes was 1.2% (95% CI: 0.7;

1.8). These results in the university population contradict the study on high-school students in the United States, where a study found that the highest prevalence was reported in the use of e-cigarettes only (18%), when compared to tobacco (only) use (3%) and dual use of e-cigarettes and tobacco (13%) [34]. These differences could be due to the wrong perception that e-cigarettes are healthier than tobacco, [41] as well as to the varying regulations and access to e-cigarettes, tobacco, and cannabis in each country [6]. Additionally, sex, ethnic and/or cultural patterns could have been an influence [40].

Even though this study has some strengths as it is one of the few studies in Latin America that assesses the use of e-cigarettes and/or tobacco in university students, is a secondary analysis of a probabilistic study with representativeness of students from ten Peruvian universities and has used measurement tools with international standards and procedures that allow to report anonymously. However, the study has some limitations as well, such as being a secondary analytical cross-sectional study and not being able to attribute a direct causal relationship since it does not include progression over time, only association. The original study assessed substance use by self-report, so there might be recall and desirability bias; moreover, it only evaluated the use of e-cigarettes in life, but neither the frequency in the last 12 months, last month or during the week nor the age of initiation of e-cigarettes use. It not either included questions to estimate sensation-seeking behaviors or the use of e-cigarettes in peers and/or parents. Besides, it did not collect any information about the sequence of use of e-cigarettes and/or tobacco or whether they were consumed simultaneously, and did not specified the components or levels of nicotine or cannabis products used as inputs in e-cigarettes, if one type was used exclusively or if they could have been used alternately.

## Conclusions

We found an increase in the probability of having used cannabis in the last 12 months among university students who reported tobacco and/or e-cigarettes use. A sequential probability gradient was found and it showed that, in comparison to students who informed being nonusers of tobacco and/or e-cigarette, university students who were lifetime dual users were most likely to having used cannabis in the last 12 months, followed by those reporting an exclusive lifetime use of tobacco, as well as those students who reported an exclusive lifetime use of e-cigarettes. It is necessary to raise awareness of the association of tobacco and e-cigarette use with an increased possibility of using cannabis. Studies are needed in different cultural contexts to investigate the progression of electronic cigarette and/or tobacco use, as well as their dosage, intensity of use, concurrent use, and neurological and behavioral mechanisms that are related to the use of cannabis and other illegal drugs that may increase dependence and mental health problems.

## Supporting information

**S1 Database. Database of the study.**
(DTA)

**S1 File. Codebook of database.**
(LOG)

## Acknowledgments

The authors thank Areli Garcia Valenzuela, who was part of the research team and greatly contributed to this study.

## Author Contributions

**Conceptualization:** Lucero Sangster-Carrasco.

**Data curation:** Lucero Sangster-Carrasco.

**Formal analysis:** Lucero Sangster-Carrasco, Dora Blitchtein-Winicki.

**Investigation:** Lucero Sangster-Carrasco, Dora Blitchtein-Winicki.

**Methodology:** Lucero Sangster-Carrasco, Dora Blitchtein-Winicki.

**Supervision:** Dora Blitchtein-Winicki.

**Validation:** Dora Blitchtein-Winicki.

**Writing – original draft:** Lucero Sangster-Carrasco, Dora Blitchtein-Winicki.

**Writing – review & editing:** Lucero Sangster-Carrasco, Dora Blitchtein-Winicki.

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
