## [Decision Letter · Decision Letter 0]

13 Jan 2022

PGPH-D-21-00881

E-cigarette and/or tobacco use and cannabis use among university students.

Dear Dr. Blitchtein-Winicki,

Thank you for submitting your manuscript to PLOS Global Public Health. After careful consideration, we feel that it has merit but does not fully meet PLOS Global Public Health’s publication criteria as it currently stands. Therefore, we invite you to submit a revised version of the manuscript that addresses the points raised during the review process.

We look forward to receiving your revised manuscript.

Kind regards,

Chandrashekhar T. Sreeramareddy

Guest Editor

Journal Requirements:

1. Please ensure that the Title in your manuscript file and the Title provided in your online submission form are the same.

2. Please provide separate figure files in .tif or .eps format only and ensure that all files are under our size limit of 20MB.

3. Please amend your detailed Financial Disclosure statement. If you did not receive any funding for this study, please simply state: “The authors received no specific funding for this work.”

4. Please update the completed 'Competing Interests' statement. If you have no competing interests to declare, please state: “The authors have declared that no competing interests exist.”

5. In the online submission form, you indicated that “The database of this study can be accessed through the request an acceptance of Office of United Nations Against Drug (UNODC) and the Comisión Nacional para el Desarrollo y Vida sin Drogas (DEVIDA) from Peru.”. All PLOS journals now require all data underlying the findings described in their manuscript to be freely available to other researchers, either 1. In a public repository, 2. Within the manuscript itself, or 3. Uploaded as supplementary information.

6. All main tables must be in an editable format for typesetting. Please include Tables 1, 2 and 3 in your main text as an editable table and not an image.

Additional Editor Comments (if provided):

Dear Dora,

The peer review of your manuscript is now complete. The feedback comments from three peer reviewers is enclosed on the e-mail and an attachment in pdf. Though the manuscript has merit and potential as contribution to the existing literature, there are some concerns about the presentation of the manuscript and some concerns about analyses carried to present the results. In my independent assessments, i concur with the reviewers feedback. We invite you to consider all the comments and submit a revised version.

Reviewers' comments:

Reviewer's Responses to Questions

**Comments to the Author**

1. Does this manuscript meet PLOS Global Public Health’s publication criteria? Is the manuscript technically sound, and do the data support the conclusions? The manuscript must describe methodologically and ethically rigorous research with conclusions that are appropriately drawn based on the data presented.

Reviewer #1: Yes

Reviewer #2: Yes

Reviewer #3: Partly

2. Has the statistical analysis been performed appropriately and rigorously?

Reviewer #1: Yes

Reviewer #2: Yes

Reviewer #3: No

3. Have the authors made all data underlying the findings in their manuscript fully available (please refer to the Data Availability Statement at the start of the manuscript PDF file)?

Reviewer #1: Yes

Reviewer #2: No

Reviewer #3: Yes

4. Is the manuscript presented in an intelligible fashion and written in standard English?

Reviewer #1: Yes

Reviewer #2: Yes

Reviewer #3: Yes

5. Review Comments to the Author

Reviewer #1: E-cigarette and/or tobacco use and cannabis use among university students

This an interesting which assess risk factors for NCD in southern America. It is very stricking there few studies on these risk factors and uthours have conducted a cross-sectional study amaongs younger population. It will have been more poerful is this covered adults and older popuation as a population based survey.

There few minor points.

a) It is important to clearly emphasise the nature and design of survey - which is cross sectional. Importantly cross-sectional survey only provide point prevalence rather than over time (longitudinal).

b)statistical comparison are fine - but i found some lack of clarity on how many use tobacco and proportion of those that use cannabis from the survey. Further how this proportion compares to other studies. This needs further re-reading and refinement in write up.

c) it is better to rely on point prevalence and 95% CI e.g. Wald. something authours can easily extract from Stata or R. Example on Table 2. P-value fine but do not offer any further infomation on uncertainty of estimate or how wide CI is. Same applies to reporting in text.

d) What dont we know from this study? I find the proportion rather small in terms of those who use tobacco and go on to use Cannabis although the odds are strickingly large. It is obvious that cannabis users would have smoked tobacco at some stage. I think the manuscript would be improved by drawing out the overall risk behaviour, NCDs and effects of Tobacco use in this population. Because we know little in terms of NCD burden, risk factors. What role could routinely collected data help improve further studies?

Reviewer #2: 1. Abstract is poorly written. Please elaborate your objectives with background why you want to investigate this issue.

2. All minor sections in the Abstract to be added in one sub-heading Methods.

3. Title is very confusing: in fact you aimed to study determinants of cannabis use in the last 12 month and those using e-cigarettes and tobacco consumption have very high risk factor of using cannabis.This should reflect in the title.

4. All tables have too much information on demographic and socioeconomic and behavioural characteristics of students.

5. Table 2A is misleading by providing life-time prevalence/use of tobacco, etc. Authors should have focused on current or past 12 month prevalence rate.

6. The findings are obvious that those who have smoking experience have used cannabis. No body i.e. a non-smoker will not straight away jump into cannabis smoking.

7. In statistical section authors have not provided any justification of choosing GLM over Logistic Regression model.

8. Too many covariates/confounders are used without providing any logical reasoning or in conceptual framework for inclusion.

9. Discussion section lacks reasoning about pivotal role these confounding variables have played.

Reviewer #3: Please see my document attached with an account of what I consider should be tackled in order to improve our knowledge regarding gateway theory including a different method to approach the association between e-cigarette and cannabis.

6. PLOS authors have the option to publish the peer review history of their article (what does this mean?). If published, this will include your full peer review and any attached files.

**Do you want your identity to be public for this peer review?** For information about this choice, including consent withdrawal, please see our Privacy Policy.

Reviewer #1: No

Reviewer #2: **Yes: **Anil Gumber

Reviewer #3: **Yes: **Jose Ignacio Nazif-Munoz

---

## [Decision Letter · Decision Letter 1]

17 Feb 2022

PGPH-D-21-00881R1

Lifetime e-cigarette and/or tobacco use and last year cannabis use among university students: a cross sectional secondary analysis of a National Survey

Dear Dr. Blitchtein-Winicki,

Thank you for submitting your manuscript to PLOS Global Public Health. After careful consideration, we feel that it has merit but does not fully meet PLOS Global Public Health’s publication criteria as it currently stands. Therefore, we invite you to submit a revised version of the manuscript that addresses the points raised during the review process.

We look forward to receiving your revised manuscript.

Kind regards,

Chandrashekhar T. Sreeramareddy

Guest Editor

Journal Requirements:

Additional Editor Comments (if provided):

Reviewers' comments:

Reviewer's Responses to Questions

**Comments to the Author**

1. If the authors have adequately addressed your comments raised in a previous round of review and you feel that this manuscript is now acceptable for publication, you may indicate that here to bypass the “Comments to the Author” section, enter your conflict of interest statement in the “Confidential to Editor” section, and submit your "Accept" recommendation.

Reviewer #1: All comments have been addressed

Reviewer #2: All comments have been addressed

Reviewer #3: (No Response)

2. Does this manuscript meet PLOS Global Public Health’s publication criteria? Is the manuscript technically sound, and do the data support the conclusions? The manuscript must describe methodologically and ethically rigorous research with conclusions that are appropriately drawn based on the data presented.

Reviewer #1: Yes

Reviewer #2: Partly

Reviewer #3: Partly

3. Has the statistical analysis been performed appropriately and rigorously?

Reviewer #1: Yes

Reviewer #2: Yes

Reviewer #3: Yes

4. Have the authors made all data underlying the findings in their manuscript fully available (please refer to the Data Availability Statement at the start of the manuscript PDF file)?

Reviewer #1: Yes

Reviewer #2: Yes

Reviewer #3: Yes

5. Is the manuscript presented in an intelligible fashion and written in standard English?

Reviewer #1: Yes

Reviewer #2: No

Reviewer #3: No

6. Review Comments to the Author

Reviewer #1: There was coherence in comments by authours on common themes raised across different reviewer and subsequent implementation in the manuscript. Corrections and consideration on presenation of tables has been rigorous.

There is also a clear desire to stick with original hypothesis/theory i.e. on predictable progression therory (Gateway theory) in drug use. The data deficiencies around this theory have however been outlined with the revised submission

Reviewer #2: Most of my comments are addressed. However I still need the abstracts to be re-written. For example one can't understand what you are saying in the opening sentence in the abstract. Similarly conclusion section conclude the research and provide policy implications and future direction to research on this topic and NOT the summary of results. Finally, I would add Correlates or Determinants or Factors as well as Peru in the title of the paper e.g. "Correlates of Lifetime e-cigarette and/or tobacco use and last year cannabis use among Peruvian university students: a cross sectional secondary analysis of a National Survey". Finally expand any abbreviations used in the Abstract (PRa or CI95, etc.).

Reviewer #3: Thank you for tackling most of my suggestions. I firmly believe you should drop all the theoretical aspects described in the introduction and then in the discussion. In both cases the writing is poor because the connection between theoretical ideas is not at all there. When one reads it one has the impression that authors had to for certain reason bring in some "theories" As I said in my first review the manuscript stands descriptively. Descriptive papers are also extremely valuable, particularly after the adjustment, as you did in this version, is properly justified.

7. PLOS authors have the option to publish the peer review history of their article (what does this mean?). If published, this will include your full peer review and any attached files.

**Do you want your identity to be public for this peer review?** For information about this choice, including consent withdrawal, please see our Privacy Policy.

Reviewer #1: No

Reviewer #2: **Yes: **Anil Gumber

Reviewer #3: No

---

## [Decision Letter · Decision Letter 2]

14 Apr 2022

Association of lifetime e-cigarette and/or tobacco use and last year cannabis use among university students: a cross-sectional secondary analysis of a National Survey

PGPH-D-21-00881R2

Dear MD MPH DrPH Blitchtein-Winicki,

We are pleased to inform you that your manuscript 'Association of lifetime e-cigarette and/or tobacco use and last year cannabis use among university students: a cross-sectional secondary analysis of a National Survey' has been provisionally accepted for publication in PLOS Global Public Health.

Best regards,

Chandrashekhar T. Sreeramareddy

Guest Editor

Reviewer Comments (if any, and for reference):

Reviewer's Responses to Questions

**Comments to the Author**

1. If the authors have adequately addressed your comments raised in a previous round of review and you feel that this manuscript is now acceptable for publication, you may indicate that here to bypass the “Comments to the Author” section, enter your conflict of interest statement in the “Confidential to Editor” section, and submit your "Accept" recommendation.

Reviewer #3: All comments have been addressed

2. Does this manuscript meet PLOS Global Public Health’s publication criteria? Is the manuscript technically sound, and do the data support the conclusions? The manuscript must describe methodologically and ethically rigorous research with conclusions that are appropriately drawn based on the data presented.

Reviewer #3: (No Response)

3. Has the statistical analysis been performed appropriately and rigorously?

Reviewer #3: Yes

4. Have the authors made all data underlying the findings in their manuscript fully available (please refer to the Data Availability Statement at the start of the manuscript PDF file)?

Reviewer #3: No

5. Is the manuscript presented in an intelligible fashion and written in standard English?

Reviewer #3: Yes

6. Review Comments to the Author

Reviewer #3: Recommendations were addressed

7. PLOS authors have the option to publish the peer review history of their article (what does this mean?). If published, this will include your full peer review and any attached files.

**Do you want your identity to be public for this peer review?** For information about this choice, including consent withdrawal, please see our Privacy Policy.

Reviewer #3: No
